# Intrastromal graft of anterior lens capsule. A substitute for Bowman layer graft transplantation for keratoconus

**Carlos A. Rodríguez-Barrientos**[1,2]*, **Amir Translateur-Grynspan**[1], **Judith Zavala**[2], **Jorge E. Valdez**[2], **Gisella Santaella**[1], **Carmen Barraquer-Coll**[1]

**1** Escuela Superior de Oftalmologia, Instituto Barraquer de America, Bogota, Colombia, **2** Escuela de Medicina y Ciencias de la Salud, Tecnologico de Monterrey, Monterrey, Mexico

* c.rodriguezb@tec.mx

## Abstract

### Purpose

The shortage of donor corneas limits Bowman layer transplantation for keratoconus. In this study, we evaluate the clinical outcome of porcine anterior lens capsule (xenograft) transplantation in the corneal stroma of a rabbit model as substitute for Bowman layer graft that is used in stromal transplantation for advanced keratoconus.

### Methods

Transplantation of porcine anterior lens capsule in the corneal stroma was performed in four New Zealand white rabbits through the creation of a stromal pocket. Corneal transparency, central corneal thickness, and topographic characteristics of corneas were evaluated at different time points: pre- (0) and post- (7, 14, 21, and 28) operative days. Additionally, at the end of the study histopathological findings were evaluated.

### Results

In comparison to pre-operative day, transplantation of an anterior lens capsule preserved corneal transparency, central corneal thickness, and topographic characteristics remained constant throughout the study period. Histopathological analysis revealed the presence of the anterior lens capsule as a fully integrated lamellar graft without adverse effects in host stroma.

### Conclusion

The anterior lens capsule may be useful as a graft for intrastromal corneal trasplantation. Similarly to Bowman layer, anterior lens capsule has mechanical characteristics that facilitate corneal transplantation. In post-transplanted corneas the preservation of transparency, as well as the effect on corneal thickness, and topographic characteristics support the possibility of using anterior lens capsule as a substitute for Bowman layer graft.

**Data Availability Statement:** All relevant data are within the paper and its Supporting Information files.

**Funding:** This work was supported by Fondo de Investigación-FI Escuela Superior de Oftalmología, Instituto Barraquer de America. Convocatoria Dr. Francisco Barraquer Coll.

**Competing interests:** The authors have declared that no competing interests exist.

## Introduction

Keratoconus (KC) is a disorder characterized by progressive ectasia with compromised optical function [1]. Different treatment options for KC have been established. These are corneal crosslinking, and intracorneal ring segments for mild to moderate KC, and penetrating kerato-plasty (PK) or deep anterior lamellar keratoplasty (DALK) for the more advanced grade of KC [2]. They are associated with well-known post-operative complications such as abnormal wound healing, suture-related problems, allograft rejection, and ophthalmological complications associated with steroid therapy [3].

Transplantation of the Bowman layer (BL) in the corneal stroma is a recent surgical approach that has demonstrated stabilization of ectasia in patients with advanced KC [4]. This approach has been developed in response to the observation that the BL of corneas with KC shows areas of fragmentation [5]. The technique consists in the transplantation of an isolated BL graft into a stromal pocket of corneas with ectasia and has gained interest as it postpones other surgical treatments such as PK or DALK [4].

BL transplantation has other potential advantages including: i) a low risk for graft rejection because the BL is an acellular membrane, ii) has less suture-related problems, and iii) corti-coid-related problems are diminished since it is an acellular membrane and lower doses of corticosteroids are used [6]. The widespread acceptance of this procedure may be limited by the current shortage of available isolated BL tissue for transplantation. Usually, the graft can be isolated from corneas used for the preparation of endothelial grafts, and donor corneas that are not accepted for transplantation [7]. This corneal dependence requires persistent efforts to overcome the ever-increasing demand for donor corneal tissues.

Technically, the preparation of a BL graft is a relatively new practice, it is a task that requires experience to avoid tearing the layer, which would make it unsuitable for transplantation. The relatively high failure rate when compared, for example, with Descemet Membrane Endothelial Keratoplasty (DMEK) graft preparation demonstrated the complexity of the procedure. Despite the promising results, donor-dependency, a delicate preparation technique, and technical challenges associated with the procedure have resulted in limited acceptance [7].

The need to overcome these limitations points toward further efforts to identify a novel source of graft that is phenotypically as close as possible to BL graft, and that should, in addition, be preferentially isolated from an ocular source obtained from an eye bank, abundant, and easily prepared.

BL and anterior lens capsule (ALC) are thick transparent ocular membranes that have in common the presence of collagen, which provides them with rigidity [8]. The BL is primarily composed of collagen types I, III, V, and VII [9], and its thickness is approximately 16–21 μm in normal eyes [10]. In contrast, the ALC is the thickest basement membrane of the body, measuring 25–30 μm [11], one of its major components is collagen type IV [12] which interacts with collagen XV, XVIII, perlecan and laminin to form a three-dimensional matrix [11].

The mechanical functions of the ALC are suspension and accommodation of crystalline lens [13]. The precise physiological function of BL is unknown and not completely understood. Recent biomechanical studies have failed to demonstrate the contribution of BL to the mechanical stability of the cornea [14]. However, surgical techniques such as BL transplantation suggest an important role in corneal biomechanics since the transplantation of BL reduces ectasia in eyes with advanced KC [15].

We hypothesize that ALC shares key characteristics with BL and could be considered as a substitute for transplantation in the corneal stroma and may offer the possibility of obtaining similar results to the current strategy of BL transplantation using the pocket technique [4]. To

our knowledge, this is the first experimental transplantation of ALC as a substitute for BL graft that is used in the corneal stroma for KC.

## Materials and methods

### Animals

Four male, *oryctolagus*, *cuniculus*, New Zealand white rabbits (2–4 months of age, and 2–3 kg of body weight) were used in the experiments. They were kept at constant temperature (22 ± 1˚C) and humidity (60 ± 10%), with a 12:12 hours light-dark cycle, and unrestricted access to standard diet and water. All animal procedures were approved by the Ethics Committee of the Escuela Superior de Oftalmología-Instituto Barraquer de América (No. C-20210714-1) and followed the ARVO Statements for the Use of Animals in Ophthalmology and Vision Research.

### Preparation of ALC graft

Four fresh porcine eyes were purchased from a local butcher, a few hours after animals were killed at a local abattoir. The eyes were disinfected by immersion in a povidone-iodine solution 5% (Oftalmo-Quimica S.A, Bogota, Colombia) for 2–5 min and then washed with balanced salt solution (BSS, Corpaul, Antioquia, Colombia). Corneoscleral buttons with the iris and the attached crystalline lens were procured by an ophthalmologist within 24 hours after porcine donor death. The corneoscleral buttons and the iris were removed for isolation of the crystalline lens, whilst keeping the localization of ALC. The lens was stained with trypan blue dye (Tecno Blue, Gujarat, India), and the ALC was isolated from the bulk of the lens by carefully cutting the periphery of the lens, detached from the lens bulk with a cannula (Fig 1A), and followed by 8–0 mm trephination (Fig 1B). Due to the inherent elasticity, the ALC graft tends to curl into a single or double roll (Fig 1C). Posteriorly, the grafts of ALC were treated with 70% alcohol for 5 minutes to debride lens epithelial cells. Finally, the samples were gently rinsed and stored in BSS until use.

### Surgical procedure

All animals were anesthetized with an intramuscular injection of a mixture of ketamine hydrochloride 35 mg/kg (Sicmafarma, Bogota, Colombia) and xylazine hydrochloride 5 mg/kg (Erma, Bogota, Colombia). The procedure was carried out in the right eye of each animal under aseptic conditions: the palpebral region was disinfected with povidone-iodine 5% and rinsed with BSS. Eyes were topically anesthetized with a drop of proparacaine hydrochloride ophthalmic solution 0.5% (ALCON, Texas, USA), and a wire lid speculum (Rumex, USA) was inserted to retract the lids for corneal exposure. An ophthalmic surgical microscope (Zeiss, Jena, Germany) was used to perform the surgical procedure. The first step was a superior conjunctival peritomy. Then 3–4 mm outside the limbus, a partial thickness scleral tunnel was made and dissected up into the clear cornea using a crescent knife (MANI, Tochigi, Japan). After this step, a circular demarcation of 8 mm vertically and horizontally was performed in the surface of the central cornea. Then, a stromal pocked was manually created with Melles´s spatula (DORC, New Hampshire, USA) over 360˚ guided by previous demarcation (Fig 1D). Once this had been accomplished, a paracentesis to the anterior chamber was created to reduce the intraocular pressure and facilitate the next steps. At the same time, the ALC graft was again washed with BSS to remove all remnant cellular material, stained with trypan blue for 30 seconds, placed on a spatula as a double roll, and then pushed into the stromal pocket with the help of a cannula (Fig 1E). Once the graft was inside the pocket, it was unfolded and positioned

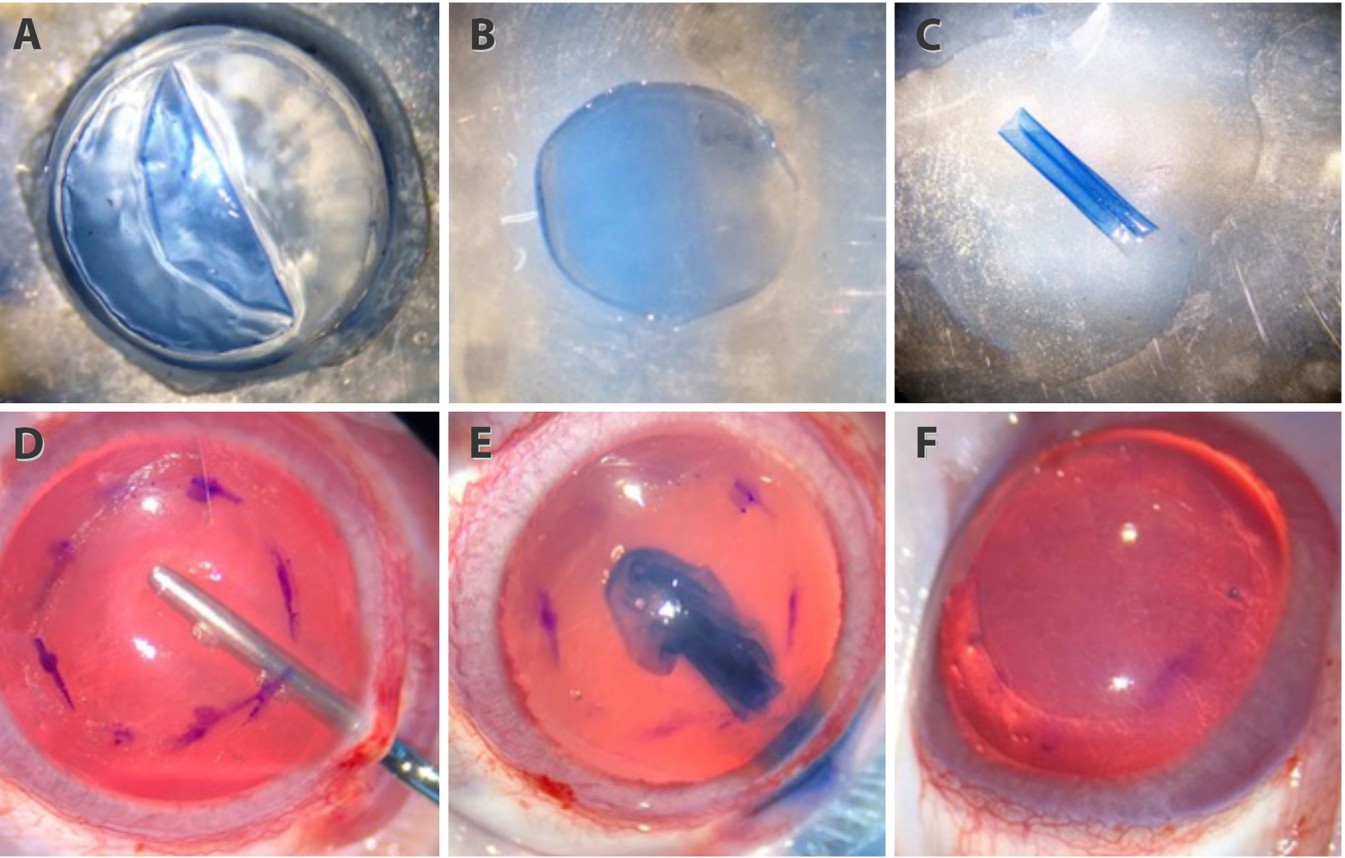

**Fig 1. Preparation and transplantation of ALC graft.** Isolation of ALC (A), 8 mm trephination (B), and doble roll configuration (C). Manually dissected stromal pocket (D), ALC graft inside of the stromal pocket (E) and unfolded and positioned in the host cornea (F).

on the center of the cornea by manipulating it with a cannula (Fig 1F). Finally, after a complete unfolding and positioning of the graft, the anterior chamber was filled with BSS, and the conjunctiva was repositioned to cover the site of the scleral tunnel. Post-operative medication included antibiotic eye drops (tobramycin 3mg/ml, Sophia, Jalisco, Mexico), 2 times daily for 10 days and corticosteroid (prednisolone acetate 1%, Sophia, Jalisco, Mexico), 3 times daily the first 7 days and then 1 time daily for one month.

## Pre- and post-operative evaluation

Each right eye was examined in pre- (0) and post- (7, 14, 21, and 28) operative days by an operator with experience for evaluation of: i) corneal transparency, ii) central corneal thickness (CCT), and iii) topographic characteristics: K-readings (K1, K2, and Km), and sagittal curvature map of the anterior corneal surface.

## Corneal transparency

Using a slit lamp biomicroscopy (Haag-Streit Diagnostics, Köniz, Suiza) corneal transparency was examined and photographed with an attached camera (Nikon D610, Japan) under standard conditions (moderate intensity, low magnification, and the room lights off). The amber filter was used for: i) a general view with diffuse illumination, and ii) an optical section of the cornea (2 mm width, 14 mm long, and 90° angle) with direct illumination.

## Central corneal thickness

An optical coherence tomography (Cirrus HD-OCT, Zeiss, Jena, Germany) was used to measure the CCT. Vertical and horizontal readings were taken using the pupil as a reference, images were captured using the automatic mode and CCT was measured at: the apex, 0–2 mm, and 2–4 mm central areas of the cornea and expressed in μm.

## Topographic characteristics

An Scheimpflug-based corneal tomography (Oculus Pentacam HR, Wetzlar, Germany) was used to measure K-readings (K1, K2, and Km) and sagittal curvature map of anterior corneal surface; images were taken using the automatic-mode and expressed in diopters (D).

## Animal tissue

On day 28, all animals were euthanized under anesthesia with an intramuscular injection of a mixture of ketamine hydrochloride 35 mg/kg (Sicmafarma, Bogota, Colombia), and xylazine hydrochloride 5 mg/kg (Erma, Bogota, Colombia), and then an intravenous injection of a mixture of sodium pentobarbital 75 mg/kg and sodium diphenylhydantoin 20 mg/kg (Invet S.A, Bogota, Colombia). Both eyes were enucleated, and corneas were then carefully excised, fixed in formol (PanReact Applicham, Darmstadt, Germany), and embedded in paraffin (Proquilab, Bogota, Colombia). Sections of 5 μm were cut with a microtome (AO Rotary microtome 820, Buffalo, USA), and stained with periodic acid-Schiff (PAS, Tacnosar, Bogota, Colombia). The left cornea was used as normal control.

## Statistical analysis

In this case, the structure of the data extracted from this experiment includes a time factor, since the measurements of the dependent variable are made in five time periods. In the search to study the effect of ALC graft on the CCT and the K-readings (K1, K2, and Km) on posttransplanted corneas, a linear model of repeated measures was applied, with the F test statistic of the ANOVA. Additionally, the Bonferroni post-hoc test is used to assess the statistical significance of the measurements to the different treatment combination. The distributional assumptions on the model are validated, with normal, homoscedastic and independent residuals [16].

# Results

For ALC graft preparation, a manual dissection from the crystalline lens was shown to be an easy technique for the obtention of a graft layer (Fig 1A) compared with the description for BL graft preparation [7]. After trephination (Fig 1B), owing to the elastic properties of ALC, a roll was formed spontaneously (Fig 1C). Similarly to the BL graft, this allowed graft manipulation for the reproduction of the transplantation of the BL graft approach. Additionally, throughout the study period, no intraoperative complications related to stromal dissections or implant of ALC graft were observed.

## Corneal transparency

In pre-operative evaluation, slit lamp biomicroscopy in all animals revealed a completely transparent and thin cornea that permitted the visualization of fine iris details and pupil in the general view image (Fig 2A), and in the optical section (Fig 2C). During the 28-day follow-up period, the exploration revealed maintenance of the corneal transparency (Fig 2B). Also, in the optical section view, post-operated cornea retained its thinness, and the ALC graft is a vaguely

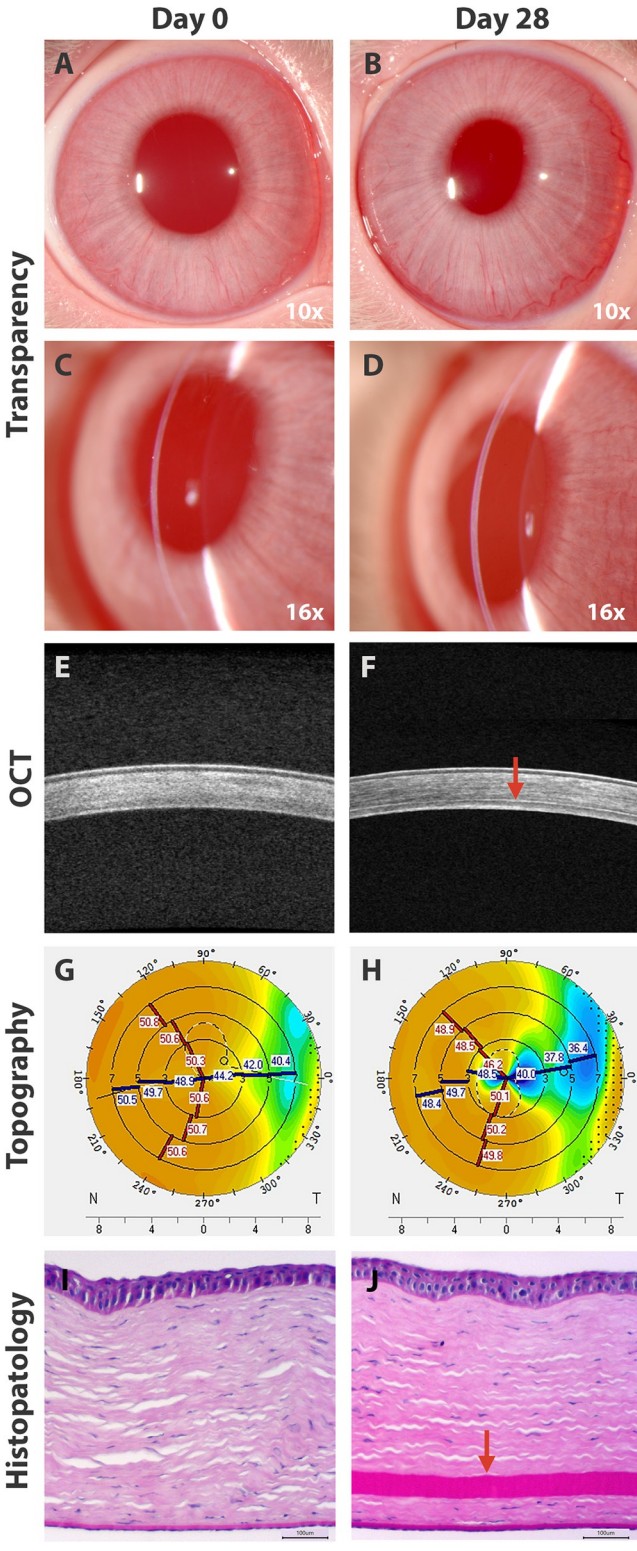

**Fig 2. Cornea pre- and post-transplantation of ALC graft.** Clinical evaluation at day 0 (A, C) and at day 28 (B, D). OCT images of pre-operative (E) and post-transplanted cornea (F), ALC graft is observed as a thin line (arrowhead). Topographic maps at day 28 (H) and day 0 (G). Histopathological features with PAS staining of pre- (I) and post-operated cornea (J) with the ALC graft (arrowhead).

and almost imperceptible thin line (Fig 2D). In 7, 14, 21, and 28 post-operative days, all transplanted corneas maintained their transparency and thickness without post-operative complications.

## Central corneal thickness

Compared with the pre-operative cornea (Fig 2E), images of OCT revealed that the ALC graft in post-operated corneas (Fig 2F) is observed as a thin, continuous line in the deep stroma. It can be seen as a change in the optical intensity sandwiched between the stromal layer above and below. There was no evidence of pathological changes in the area during the post-operative examinations. Data analysis showed pre-operative mean CCT: i) 340 ± 26.4 μm in the apex, 340 ± 27.7 μm in the central 0–2 mm area, and 336 ± 26 μm in the central 2–5 mm area. In contrast with post-operative 28 days where the mean CCT values were: i) 357 ± 27.1 μm in the apex, 358 ± 28.5 μm in the central 0–2 mm area, and 356 ± 25.6 μm in the central 2–5 mm area. Quantitative analysis reveals that CCT values were increased in post-operative days in comparison with pre-operative days, but with no statistically significant differences between the groups, with a confidence level of 99% (Fig 3).

## Topographic characteristics

Compared with pre-operative (Fig 2G) measurements of the anterior corneal surface a flattening effect was observed in all post-transplanted corneas (Fig 2H). Data analysis of K readings of the anterior corneal surface at day 0 showed that mean K1 was 48.8 ± 2.2 D, K2 was 50.6 ± 0.85 D, and Km was 49.6 ± 1.53 D. In contrast, post-operative day 28 where mean values were: K1 was 45.9 ± 4.27 D, K2 was 49.1 ± 1.43, and Km was 47.4 ± 2.89 D. The K readings values showed a regression at day 21 before stabilizing at day 28, similar to those at day 14. Quantitative analysis revealed that K readings values were reduced in post-operative days in comparison with pre-operative days, but no statistically significant differences between the groups were observed, with a confidence level of 99% (Fig 4). The flattening effect is observed

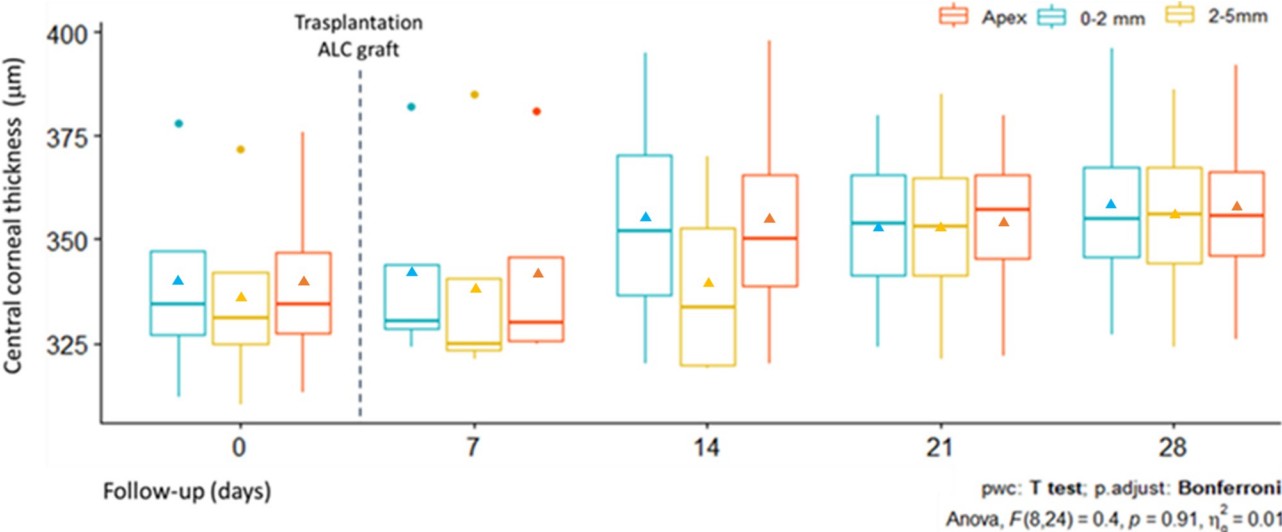

**Fig 3. Evolution of CCT after transplantation of ALC graft.** Graphs showing the increase in CCT in the apex, area of 0–2 mm, and the area of 2–5 mm in central cornea in the follow-up period. In the box plots, triangles indicate mean values.

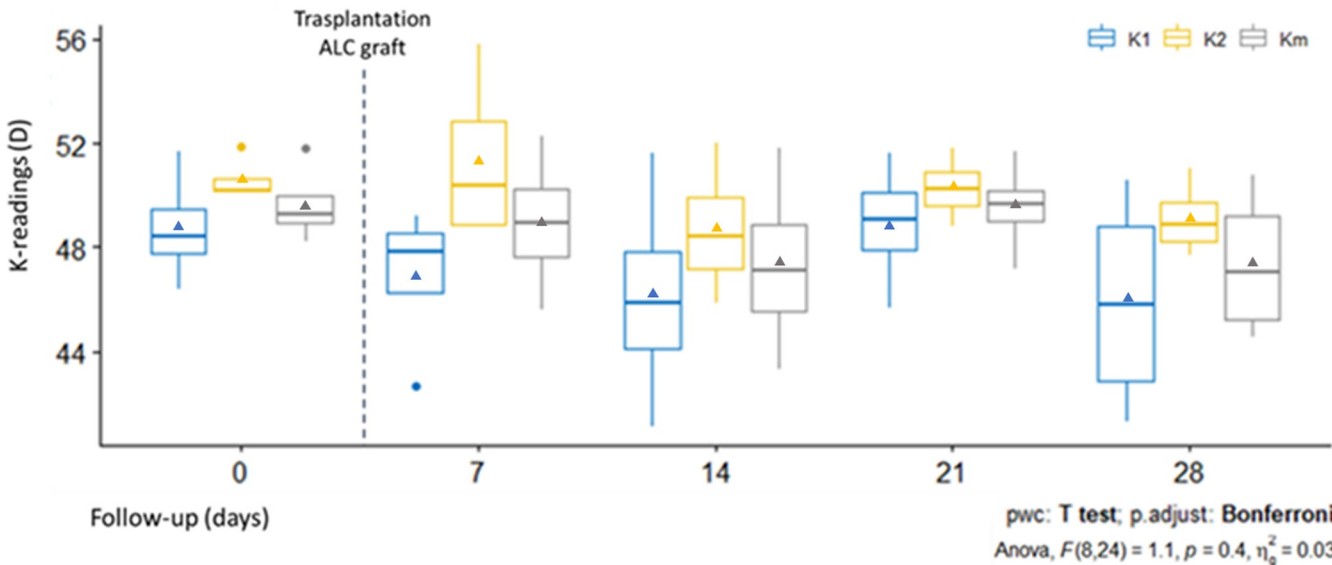

**Fig 4. Evolution of K-readings after transplantation of ALC graft.** Graphs showing the post-operative flattening in the follow-up period. In the box plots, triangles indicate mean values.

in the sagittal curvature map at the end of the study (Fig 2H) when is compared with pre-operative day (Fig 2G).

## Histological analysis

The post-transplanted corneas (Fig 2J) showed that the ALC graft was attached and integrated into the corneal stroma without pathological changes, displaying a similar histological structure when compared with the healthy contralateral cornea (Fig 2I), except for the presence of the ALC graft in the stroma.

## Discussion

BL transplantation has gained popularity as it postpones other surgical procedures as PK or DALK in patients with advanced KC since it has demonstrated stabilization of corneas with ectasia [4]. The main drawbacks of this approach are: i) the manual dissection, which is laborious and technically demanding task, and more importantly ii) the dependence on corneal donor tissue, which means the continuation of well-known efforts to overcome the lack of donor corneas.

The objective of transplanted BL graft is to restore the fragmentation of BL observed in KC [5]. We hypothesized that the use of ALC as a substitute of BL graft has the potential to produce similar effects of BL transplantation due to the similarities between both collagen membranes. This strategy overcomes the limitations associated with the preparation of BL graft.

The first outcome of this study was that the preparation of the ALC graft involves a simple and faster learning curve in comparison with the BL graft, considered a complex procedure with a relatively high failure reported around 30% [7]. The tearing of ALC during preparation is a risk, but the graft isolation after trypan blue staining with surgical scissors in the periphery of the lens, and posterior trephination results in an adequate and safe technique for graft layer preparation. In our study no graft tearing complication was presented.

Additionally, the poor attachment of the ALC to the bulk of the lens, and the easy visualization of the ALC border facilitates the obtention of only the ALC with less controlled manual

force, and without using special instruments. A cannula is sufficient to detach the ALC from the bulk of the lens, in contrast with the BL graft preparation, that according with Groeneveld-Van Beek is a task that requires sufficient experience to identify subtle differences in corneal anatomy and manual forces [7]. Additionally, BL grafts sometimes are isolated with adjacent stromal tissue associated with the vague border between the BL and stroma, which is difficult to see; even with the use of femtosecond for BL graft preparation some amounts of stroma can be isolated [17], these facts reflect the complexity of the pure BL graft preparation.

After obtention of the ALC graft, owing to its elastic properties a roll was formed spontaneously similar to the BL graft. This characteristic allows the transplantation of ALC following the current surgical procedure described by Korine van Dijk et al [4, 15]. In the surgical technique, because ALC is an acellular and elastic membrane it may be subjected to direct manipulation; similar to BL graft [4], the ALC was inserted into the stromal pocket, unfolded, and centered in the host stroma. Our results showed that the approach of insertion of ALC graft inside corneal stroma is effective, reproducible, and safe since no complications during the surgical procedure were presented.

In our study, we established a 28-day follow-up period. It was observed that transplanted corneas with ALC graft maintain their transparency throughout this follow-up period. Similar to BL graft [18], the use of ALC as an acellular graft potentially avoids the graft related complications. No evidence of pathological corneal changes was evidenced in this study, even when performing a xenograft (porcine ALC graft in a rabbit host cornea). These results are corroborated with the post-transplanted OCT images that show a thin cornea with the ALC sandwiched between anterior and posterior host stroma, and a similar average thickness is observed when compared with the pre-operative day and contralateral untouched cornea. Similar results are observed in the results of CCT of transplanted corneas with BL grafts [4].

Topographic analysis results in non-significant flattening in all post-transplanted corneas in the anterior corneal surfaces, in comparison with post-transplanted corneas with BL graft the flattening effect in the anterior surfaces is less notorious; it seems that significant corneal flattening effect is observed in more advanced cases of KC [15]. A similar result has demonstrated for other treatments in corneas with ectasia, for example in crosslinking the notable flattening effect was obtained in cases with the advanced KC [19, 20]. Since our study ALC transplantation was performed in corneas without ectasia, it is possible that this could explain the non-significant flattening results. A temporal regression after transplantation was observed in our study at day 21, and a similar phenomenon was observed in post-transplanted BL graft [15].

In our histologic analysis ALC graft appears as a fully integrated component in corneal tissue without pathological changes. To our knowledge there are no histopathological studies of post-transplanted human corneas with BL grafts to compare with our study. These results confirm that the ALC graft transplantation maintains corneal transparency, CCT, and topographic characteristics of transplanted corneas. ALC grafts present several advantages: i) it does not depend on donor corneal tissue, ii) the possibility of easy and biologically safe access from an eye bank, since the crystalline lens is discarded after procurement of the cornea and serologic testing, iii) simple technique for ALC graft preparation, and iv) adequate mechanical properties indicating that it can be used in stromal transplantation. Thus, an important advantage is that the ALC graft may be prepared at any moment, so that this graft would be more readily available for transplantation, unlike BL graft using corneoscleral buttons in which DMEK graft needs to be prepared before the obtention of BL graft [7].

The study still holds some limitations. On one hand, the number of animals were limited, and on the other a longer follow-up period will be required. Additionally, a transplant of an ALC graft in a rabbit animal model of corneas with ectasia will be necessary to elucidate the

flattening effect and the reduction of progress in pathologic cornea. However, it's important to consider that the stromal dissection in corneas with ectasia may be challenging because these corneas are very thin and fragile, with an increasing risk of complications such as perforation into the anterior chamber during surgery.

With this strategy, from one donor of whole-globe enucleation the anterior and posterior parts of the corneas could be used for different lamellar keratoplasty techniques and of course BL graft, but the possibility of obtention of ALC graft as BL graft substitute increases the donor tissue pool obtained from one eye donor. Nevertheless, in many countries *in situ* extraction of donor cornea is a preferred technique, which may limit the possibility for obtention of ALC with the proposed technique. To solve this problem, should it be possible to obtain ALC graft for capsulorrhexis cut by a femtosecond laser during femtosecond laser-assisted cataract surgery, in this scenario there would be a considerably larger pool of ALC donor.

Since capsulorrhexis diameter by femtosecond is around 5 to 6 mm, the donor size of ALC graft would not be as large as used in this study. But at this point, the minimum diameter of BL graft to stop the progression of keratoconus is unknown. It is possible that smaller graft positioned at the thinnest point of ectasia could be enough to stop the progression of disease. Therefore, further research is needed to address this aspect.

This study is the first reported use of ALC for transplantation in corneal stroma as a substitute for BL graft. The alternative to using an ocular tissue that naturally expresses a key characteristic of BL and that came from an eye source that is regularly discarded in eye banks after procurement of corneas from healthy donors, provides a novel strategy in the field.

## Conclusion

The work shows that it is possible to easily obtain an acellular ALC graft with mechanical properties that allow its intrastromal transplantation in a rabbit animal model. After the procedure corneal transparency is maintained, and the thickness and topographic characteristics of the cornea are constant. Also, stromal integration and no graft rejection in post-transplanted corneas was observed. With this approach it could be possible to use ALC as an adequate alternative to BL graft for intrastromal transplantation. Furthermore, with this strategy it is possible to enhance the use of donated eye tissue for new treatments.

## Supporting information

**S1 Table. Changes in central corneal thickness (CCT) following the transplantation of an anterior lens capsule (ALC) graft during the follow-up period.**
(PDF)

**S2 Table. Changes in K-readings following the transplantation of an anterior lens capsule (ALC) graft during the follow-up period.**
(PDF)

## Acknowledgments

The authors thank the diagnostic team from Clinica Barraquer: OD. Jannis Amaya, PH. Alberto Pinilla; Department of Research and Innovation: COORD. Maria Obdulia Jimenez; Experimental Surgery: LPN. Jeannette Arevalo Barrantes, and MD. Andres Novoa, resident of Ophthalmology Escuela Superior de Oftalmologia, Instituto Barraquer de America.

## Author Contributions

**Conceptualization:** Carlos A. Rodríguez-Barrientos, Gisella Santaella, Carmen Barraquer-Coll.

**Data curation:** Carlos A. Rodríguez-Barrientos.

**Formal analysis:** Carlos A. Rodríguez-Barrientos.

**Funding acquisition:** Carmen Barraquer-Coll.

**Investigation:** Carlos A. Rodríguez-Barrientos, Carmen Barraquer-Coll.

**Methodology:** Carlos A. Rodríguez-Barrientos, Gisella Santaella, Carmen Barraquer-Coll.

**Project administration:** Carlos A. Rodríguez-Barrientos.

**Resources:** Carlos A. Rodríguez-Barrientos, Carmen Barraquer-Coll.

**Software:** Carlos A. Rodríguez-Barrientos.

**Supervision:** Carlos A. Rodríguez-Barrientos, Carmen Barraquer-Coll.

**Validation:** Carlos A. Rodríguez-Barrientos, Carmen Barraquer-Coll.

**Visualization:** Carlos A. Rodríguez-Barrientos, Carmen Barraquer-Coll.

**Writing – original draft:** Carlos A. Rodríguez-Barrientos, Amir Translateur-Grynspan, Judith Zavala, Jorge E. Valdez, Gisella Santaella, Carmen Barraquer-Coll.

**Writing – review & editing:** Carlos A. Rodríguez-Barrientos, Amir Translateur-Grynspan, Judith Zavala, Jorge E. Valdez, Gisella Santaella, Carmen Barraquer-Coll.

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
