## [Decision Letter · Decision Letter 0]

1 Aug 2024

PONE-D-24-23473Intrastromal graft of anterior lens capsule. A substitute for Bowman layer graft transplantation for keratoconus.PLOS ONE

Dear Dr. Rodríguez,

Thank you for submitting your manuscript to PLOS ONE. After careful consideration, we feel that it has merit but does not fully meet PLOS ONE’s publication criteria as it currently stands. Therefore, we invite you to submit a revised version of the manuscript that addresses the points raised during the review process. Specifically, both reviewers made positive comments about the present manuscript. However, Reviewer 2 suggested that the author explain in the abstract and in the methods that the anterior lens capsule is taken from porcine eyes, as well as the time of taking grafts from the donors. Moreover, the viability of taking the ALC from patients undergoing conventional or femtocataract surgery should be commented. Finally, the donor size should be discussed according to the reviewer's comments.

We look forward to receiving your revised manuscript.

Kind regards,

Georgios Labiris, MD, PhD

Academic Editor

PLOS ONE

Journal Requirements:

3. Thank you for stating the following financial disclosure: "This work was supported by Fondo de Investigación-FI Escuela Superior de Oftalmología, Instituto Barraquer de America. Convocatoria Dr. Francisco Barraquer Coll."

Reviewers' comments:

Reviewer's Responses to Questions

**Comments to the Author**

1. Is the manuscript technically sound, and do the data support the conclusions?

Reviewer #1: Yes

Reviewer #2: Yes

2. Has the statistical analysis been performed appropriately and rigorously? 

Reviewer #1: Yes

Reviewer #2: I Don't Know

3. Have the authors made all data underlying the findings in their manuscript fully available?

Reviewer #1: Yes

Reviewer #2: Yes

4. Is the manuscript presented in an intelligible fashion and written in standard English?

Reviewer #1: Yes

Reviewer #2: Yes

5. Review Comments to the Author

Reviewer #1: A nice research idea for treating advanced keratoconus. Great documentation in their pilot study. I wish to complement the authors for their novel idea and a good scientific research which can go a long way in treating corneal ectasias. It is a good option especially when donor corneas are scarce and also the surgical preparation of the implant is easy and can be easily reproduced. Since anterior len capsule is acellular, chances of rejection is also less.

Reviewer #2: Thank you for the opportunity to review this paper.

Whether ALC can substitute Bowman's layer in stabilising eyes with keratoconus is highly speculative. It would be good if the authors could provide any further research or evidence to demonstrate that the membrane function of ALC is in any way similar to that of Bowman's layer.

Nonetheless, I believe that this paper has explained the rationale for the study, and presented the methods and results clearly. The results appear to confirm the authors hypothesis, that intrastromal transplantation of an anterior lens capsule is feasible, and at least in this rabbit model, demonstrates no adverse effects. As such, I believe the paper deserves to be accepted.

I would advise the authors to make it a little clearer in the abstract and in the methods that the ALC is taken from porcine eyes. This is not explicit in the abstract. Also, in line 125-126, I think it would help if that authors started that the corneoscleral buttons and the iris were harvested within 24 hours of the death of the "porcine donor".

The authors may also wish to comment on the fact that in many countries, whole globes are no longer taken during cornea donation, which may limit the viability for ALC to be harvested with the technique they described. Conversely, they may wish to comment if taking ALC from patients undergoing cataract surgery might be viable, in which case the pool of ALC donors would be significantly larger. ALC that has been cut by a femtosecond laser during femtocataract surgery might provide a suitable round donor for this transplant. The donor size of course would not be as large, but at this point it is not clear if ALC transplants (or even Bowman's membrane transplants) require a large 9-10 mm graft to be effective, or whether a 6mm graft centered at the thinnest point of the cone will suffice.

6. PLOS authors have the option to publish the peer review history of their article (what does this mean?). If published, this will include your full peer review and any attached files.

Reviewer #1: No

Reviewer #2: No

---

## [Author Response · Author response to Decision Letter 0]

4 Sep 2024

We would like to express our gratitude to the reviewers for their valuable comments. In response to their insightful feedback, we have made substantial revisions to enhance the quality and scientific rigor of our study. 

Reviewer 1: We are grateful to Reviewer 1 for the positive comments to our study.

Reviewer 2: We are grateful to Reviewer 2 for their very constructive feedback. In their comment regarding be clear in the abstract and in the methodology that the ALC is taken from porcine eye. We have taken the suggestion into account and included in the abstract (line 22 and 27), and in the Material and methods section, line 122 mentions “Four fresh porcine eyes…”. In this section in line 127, we mention that grafts were procured within 24 hours after porcine donor death.

Also, Reviewer 2 highlighted the need to mention the possibility of taking ALC graft from cataract patients undergoing femto cataract surgery. The recommendation has been beneficial and interesting. We have taken this suggestion into account and included in the Discussion section, lines 379 to 388. Finally, the “donor size” topic is another interesting observation mentioned by Reviewer 2, in response to this suggestion, we include this idea in the Discussion section, line 390 to 395, and in a search of this topic it is known to us that no information is reported. We believe this addition strengthens the overall scientific rigor of our work.

---

## [Editor Report · Decision Letter 1]

30 Sep 2024

Intrastromal graft of anterior lens capsule. A substitute for Bowman layer graft transplantation for keratoconus.

PONE-D-24-23473R1

Dear Dr. Rodríguez-Barrientos,

We’re pleased to inform you that your manuscript has been judged scientifically suitable for publication and will be formally accepted for publication once it meets all outstanding technical requirements.

Kind regards,

Georgios Labiris, MD, PhD

Academic Editor

PLOS ONE

Additional Editor Comments:

The authors have sufficiently answered all reviewers' comments and made the necessary corrections in the main manuscript. Moreover, this topic is timely and will be of high interest to the journal readers. As a result, this paper is appropriate for publication. Further research could contribute contemporary information to this important scientific topic.

---

## [Editor Report · Acceptance letter]

8 Nov 2024

PONE-D-24-23473R1 

PLOS ONE

Dear Dr. Rodríguez-Barrientos, 

I'm pleased to inform you that your manuscript has been deemed suitable for publication in PLOS ONE. Congratulations! Your manuscript is now being handed over to our production team.

Kind regards, 

on behalf of

Dr. Georgios Labiris 

Academic Editor

PLOS ONE